# Inoculum of Endophytic *Bacillus* spp. Stimulates Growth of Ex Vitro Acclimatised Apple Plantlets

**DOI:** 10.3390/plants14071045

**Published:** 2025-03-27

**Authors:** Jurgita Vinskienė, Inga Tamošiūnė, Elena Andriūnaitė, Dalia Gelvonauskienė, Rytis Rugienius, Muhammad Fahad Hakim, Vidmantas Stanys, Odeta Buzaitė, Danas Baniulis

**Affiliations:** 1Institute of Horticulture, Lithuanian Research Centre for Agriculture and Forestry, Kaunas Str. 30, 54333 Babtai, Kaunas reg., Lithuania; jurgita.vinskiene@lammc.lt (J.V.); inga.tamosiune@lammc.lt (I.T.); elena.andriunaite@lammc.lt (E.A.); seteksna@gmail.com (D.G.); rytis.rugienius@lammc.lt (R.R.); fahad.hakim@lammc.lt (M.F.H.); vidmantas.stanys@lammc.lt (V.S.); 2Department of Biochemistry, Vytautas Magnus University, Universiteto Str. 10, 53361 Akademija, Kaunas reg., Lithuania; odeta.buzaite@vdu.lt

**Keywords:** dormant buds, in vitro propagation, *Malus* sp., metagenomic analysis, plant growth-promoting bacteria, rooting

## Abstract

In vitro shoot culture and cryopreservation (CP) are techniques essential for the ex situ preservation of genetic resources and the production of plant propagation material of clonally propagated horticultural crops. Changes in plant-associated microbiota diversity and composition induced by in vitro cultivation and CP treatment could have a negative effect on the growth and ex vitro adaptation of the in vitro propagated shoots. Therefore, the aim of the present study was to assess changes in endophytic bacteria diversity in domestic apple tissues induced by in vitro cultivation and CP treatment and to investigate the potential of the bacterial inoculum to improve the rooting and ex vitro acclimatisation of the propagated shoots. Metataxonomic analysis revealed a variation in the endophytic bacteria diversity and taxonomic composition between the field-grown tree dormant bud and the in vitro propagated or CP-treated shoot samples of apple cv. Gala. Whereas *Sphingobacteriaceae*, *Sphingomonadaceae*, *Pseudomonadaceae*, and *Beijerinckiaceae* families were the most prevalent families in the bud samples, *Enterobacteriaceae*, *Bacillaceae*, and *Lactobacillaceae* were dominant in the in vitro shoots. The bacterial inoculum effect on rooting and ex vitro acclimatisation was assessed using four isolates selected by screening the endophytic isolate collection. *Bacillus* sp. L3.4, *B. toyonensis* Nt18, or a combined inoculum resulted in a 21%, 36%, and 59% increase in cumulative root length and a 41%, 46%, and 35% increase in the biomass accumulation of ex vitro acclimatised plantlets, respectively. Root zone microbiota functional diversity analysis implied that growth stimulation was not related to improved nutrient uptake but could involve a pathogen-suppressing effect. The results demonstrate that the application of plant growth-promoting bacteria can potentially improve the performance of the in vitro propagated germplasm.

## 1. Introduction

In vitro cultivation and cryopreservation (CP) techniques are essential to produce pathogen-free plant propagation material of horticultural crops. Since the second half of the 20th century, certification programmes have been implemented in many countries [1], and pathogen-free plant material has been widely used for vegetatively propagated horticultural crops, such as fruit trees [2,3,4], strawberry [5], or ornamentals [6]. The long-term storage of nuclear stock used for certified plant propagation requires specialised facilities with strict isolation conditions; it is also associated with the risk of material loss, and repeated quality validation is time- and cost-consuming. Hence, the application of in vitro shoot tip culture and CP techniques presents an attractive alternative for long-term virus storage and graft-transmissible pathogen-free nuclear stocks used to prepare certified plant propagation material.

The domestic apple (*Malus* × *domestica* Borkh.) is one of the most important horticultural plants in temperate regions and is one of three model species of the *Rosaceae* family [7]. The in vitro techniques have been used for the micropropagation of apple rootstocks, the propagation of new varieties, breeding lines, and disease-free fruit plants [8,9,10,11], and the regeneration of transgenic lines [12]. A review by Dubranszki et al. [13] with a subsequent update by Teixeira da Silva et al. [14] assessed more than 60 studies on optimising rooting conditions for more than 50 in vitro propagated apple genotypes. Extensive research over four decades established a two-phase procedure, including different media composition and growth condition requirements for the root induction and elongation phases. Different auxins and their concentrations, such as 0.3–3.0 mg L^–1^ indole-3-butyric acid (IBA), 0.1–1.0 mg L^–1^ α-naphthaleneacetic acid (NAA), or 0.3–1.0 mg L^–1^ indole-3-acetic acid (IAA), have been shown to be effective for the induction of roots for different genotypes [14].

Several studies also addressed ex vitro acclimatisation of the propagated apple shoots (reviewed by Dubranszki et al. [13]). Besides efforts to prevent desiccation [15] and improve the adaptation of photosynthetic apparatus [16], protecting the plantlets from bacterial and fungal diseases was an important issue addressed in several studies. An application of fungicides was suggested by Isutsa et al. [17] to improve the efficiency of apple rootstock acclimation. Modgil et al. [18] demonstrated that soilless media containing coco-peat effectively prevented disease development. Similarly, Mir et al. [18] used moist cotton for the first 10–12 days of acclimatisation, followed by a vermiculite/coco-peat mixture. Alternatively, the application of the biological plant protectant *Trichoderma harzianum* could improve plantlet survival by up to 70–100%, depending on the genotype [19]. Mycorrhization using arbuscular mycorrhizal fungi of *Glomus* and *Gigaspora* spp. was also shown to improve the performance of acclimated apple plantlets [20,21,22,23]

The potential of plant growth-promoting (PGP) bacteria inoculum to improve rooting or acclimatisation of in vitro propagated apple has not been demonstrated so far; however, the successful use of PGP bacteria for other in vitro applications has been reported for apple and other species. Bacteria that constitute the plant microbiome play an important role in plant adaptation and survival in unfavourable environments. This also applies to stress factors associated with in vitro cultivation conditions, such as suboptimal cultivation medium composition, high air humidity, low irradiance, and low CO_2_ concentration during the light period, which could lead to imbalanced physiological equilibrium and stress. It has been reported that the complementation of axenic culture with endophytic bacteria could mitigate stress and improve the growth of in vitro shoot cultures of apple [24], sweet cherry [25], tobacco [26], tomato [27], grapevine [28], and coneflower [29]. Also, the stimulation of the rooting of difficult-to-propagate sweet cherry genotypes in vitro by endophytic bacteria inoculum has been described [30,31].

Apple buds are commonly used to establish in vitro shoot culture. Previously, the endophytic microbiota of shoot and dormant bud tissues of several apple genotypes has been investigated using metagenomic analysis [32,33]. Cultivable bacteria with PGP properties were isolated from apple buds by Miliute et al. [34]. However, there is a limited understanding of in vitro cultivation and CP treatment effects on endophytic bacteria diversity. Therefore, in the present study, we investigated changes in endophytic bacteria diversity in field-grown tree dormant buds and in vitro propagated and CP-treated shoots of apple cv. Gala. Further, the screening of PGP activity of bacteria isolated from different plant sources was performed. Co-cultivation experiments were carried out to assess the capacity of the selected bacterial isolates to promote in vitro propagated apple shoot rooting and to improve growth during ex vitro acclimatisation. The bacterial inoculum effect on taxonomic and functional diversity in the roots and root zone microbiota of the acclimatised apple plantlets was assessed using metagenomic analysis.

## 2. Results

### 2.1. Apple In Vitro Shoot Culture and Cryopreservation

The apple cv. Gala in vitro shoot culture was initiated from axillary dormant buds collected in early spring just before the buds emerged. From the established explants, several stably growing mericlones were obtained after 3–6 months of cultivation, and a shoot culture with the highest growth vigour was selected for CP and microbial inoculation experiments (Figure 1A).

Cold preconditioning of in vitro shoot culture was used to improve tissue tolerance to CP treatment. Following the preconditioning treatment, the apical shoot tips were excised, treated with cryoprotectants, and subjected to cryogenic freezing. After thawing and 2 weeks of regeneration, the survival rate of apical shoot tips was 69.6 ± 8.0%. The shoot regrowth rate estimated after one month of cultivation of regenerated shoot tips was 37.2 ± 9.0% (Figure 1B).

### 2.2. Diversity of Endophytic Bacteria in Apple Bud and In Vitro Shoot Tissues

To assess the effect of in vitro cultivation and CP treatment on the endophytic bacteria community of apple tissues, metataxonomic analysis was carried out using samples of field-grown tree dormant buds and in vitro propagated shoots before and after CP treatment. High-throughput sequencing of the *16S rRNA* gene domain V4 yielded 23,569 to 128,236 high-quality reads. A large proportion (61.5–99.1%) of the amplicon sequence variants (ASVs) were either not assigned to any taxonomy or mapped to the mitochondrial *16S rRNA* gene (Appendix A). The proportion of ASVs mapped to plastidial *16S rRNA* was less than 1%, suggesting that many of such reads remained unassigned, likely due to the relatively short length (220 bp) of the reads after trimming. The proportion of bacterial ASVs varied among the experimental groups. A higher proportion of bacterial ASVs was obtained for bud samples (26–39%); meanwhile, in vitro propagated shoot and CP-treated shoot samples had a similar proportion of bacterial ASVs, ranging from 0.9% to 1.9%. A variation in the yield of bacterial ASVs was likely related to the differences in the ratio of bacterial and mitochondrial/plastidial abundance in the apple tissues [35], which was consistent with the presence of lower plastidial DNA content in predominantly non-photosynthetic tissues of the dormant buds compared to photosynthetic tissues of in vitro propagated shoots.

The principal coordinate analysis (PCoA) of the metataxonomic data revealed a significant (*p* < 0.016) difference in bacterial population composition between bud and in vitro propagated shoot or CP-treated shoot samples (Appendix A). The differences were related to bacterial diversity and taxonomic composition variation between the experimental groups. Alpha diversity estimates at the family level revealed reduced bacterial taxa richness in the in vitro culture compared to the bud samples (Figure 2; Appendix A). The effect was more pronounced in the CP-treated shoot samples. A less significant difference between the bud and in vitro shoot samples was detected using Shannon or Simpson’s diversity estimates, which implied the evenness of taxa distribution within the microbial community (Figure 2B; Appendix A). Furthermore, more prominent diversity differences between bud and in vitro shoot samples were detected using ASV-level data (Figure 2C,D), implying higher diversity in bud tissues at the genus or species level. Since taxonomic assignment below the family level is limited for short-read sequencing data, a long-read metataxonomic analysis or other approaches would be required to confirm this observation. Overall, the diversity analysis results suggested that the richness of endophytic bacterial taxa and the dominance of one or more species was considerably reduced when tissues were introduced to in vitro culture.

The bacterial ASVs were mapped to 37 families, 9 of which were present in all three experimental groups (Appendix A). Seventeen families were detected in the in vitro shoot culture samples before and after CP treatment. However, of these, only seven were detected in bud samples. The bud samples included an additional 19 unique families. Although many unique families of the in vitro shoot or bud samples had only low read abundance, several differences were prominent in the core microbiome of these tissues (Figure 3). Alpha-proteobacteria of *Sphingomonadaceae*, *Beijerinckiaceae* families, and actinobacteria of the *Microbacteriaceae* family were prevalent in the bud samples, representing, on average, 28.5, 16.4, and 6.1% of all reads, respectively. In addition, *Oxalobacteriaceae*, *Sphingobacteriaceae*, *Nocardiaceae*, and *Acetobacteraceae* were detected mainly in bud samples. Conversely, *Enterobacteriaceae*, *Bacillaceae*, and *Lactobacillaceae* were predominant in the in vitro samples, where they constituted 46.6, 13.3, and 9.6% of all reads, respectively. Only bacteria of the *Pseudomonadaceae* family were common in all experimental groups. The results highlight the dynamic transition in bacterial community structure of apple tissues introduced to in vitro culture from dormant buds. Further but less pronounced changes were introduced by the CP treatment of in vitro shoots, which led to a reduced abundance of bacteria of *Microbacteriaceae*, *Moraxellaceae*, *Mycobacteriaceae*, and *Neisseriaceae* families.

### 2.3. Screening of Plant Growth-Promoting Endophytic Bacteria

Changes in endophytic microbiota composition induced by in vitro cultivation conditions and/or CP treatment could have a negative effect on in vitro culture growth vigour. Inoculation of in vitro culture with PGP bacteria could restore or improve shoot growth [25]. In this study, we assess the PGP bacteria’s effect on apple in vitro shoot rooting and ex vitro adaptation. A collection of 104 isolates was used in the preliminary screening to select PGP bacteria for the inoculation experiments. These included 23 isolates from apple leaves [34], which represent the same species used in the inoculation experiments. However, a number of studies reported the PGP effect of endophytes on non-host plants (e.g., [37,38,39]); therefore, endophytic bacteria isolates obtained from distinct plant species, such as tobacco leaves (32 isolates) [26], *Rhodiola rosea* rhizome (40) [40], and strawberry stems (9), were also included in this study. The potential PGP properties of the isolates were previously characterised using biochemical and genetic tests or plant in vitro assays, as described in the provided references.

To colonise plant tissues, most endophytic bacteria should be able to either avoid or suppress the microbe-associated pattern-triggered response of the host plant. Since the production of reactive oxygen species is one of the earliest cellular responses following the recognition of microbe-associated molecular patterns [41], an analysis of the intracellular redox balance of plant cells could provide insights into the capacity of the microorganisms to engage in endophytic interaction with plant cells. A tobacco cell culture-based screening system for microorganisms that activate plant immune response was previously described [42]. In our study, a similar approach was used as a first step to select candidates for further analysis of PGP properties. The bacterial isolates were co-cultivated with tobacco plant cell suspension culture, and the effect on the intracellular redox balance of tobacco cells was assessed (Figure 4). Approximately 24% of isolates had strong reactive oxygen species production-suppressing activity, which indicates a strong defence response leading to the death of the tobacco cells. This effect was more prevalent among the bacteria of class Gammaproteobacteria. Contrarily, most of the bacteria of class Bacilli induced low or moderate changes in the cell redox balance compared to the control. Although the defence response is often host plant-specific and might vary for different plant species, the isolates that did not trigger or suppress immediate defence response would be more likely to establish compatible interaction with the plant cells.

Further, six isolates of *Bacillus* spp. and one each of the *Peribacillus*, *Pseudomonas*, *Rhodococcus*, *Serratia*, and *Xanthomonas* genera (as indicated in Figure 4) were selected for analysis of tobacco plant growth regulation in vitro. The selected isolates were co-cultivated with germinating tobacco seedlings, and radicle- and hypocotyl-growth-modulating effects were assayed (Figure 5; Appendix A). All isolates stimulated radicle growth 8–26%, whereas the impact on seedling shoots was less pronounced, and only seven isolates significantly stimulated hypocotyl growth 5–14%. *Bacillus toyonensis* Nt18 had the most substantial growth-stimulating effect on both radicle and hypocotyl, resulting in a 17 ± 1.3% and 14 ± 1.6% increase in length compared to the control.

In addition, the indole-related compound (IRC) producing activity of endophytic bacteria isolates was assessed using an in vitro assay. Five of the tested isolates showed the production of significant amounts of IRC (Figure 6). Interestingly, contrasting results were observed for the three *B. toyonensis* isolates. High levels of IRC production by *B. toyonensis* S1.4 or *S. fonticola* Ber.2.1 could be related to the tobacco seedling root growth-stimulating activity of the two isolates (Figure 5). An improved rooting efficiency of in vitro propagated apple shoots using natural IAA compounds produced by bacteria was previously reported [43]. In our study, the application of IRC-producing endophytic isolates could potentially complement or replace the root-inducing activity of the IAA in the apple in vitro shoot rooting medium. However, the interpretation of the cumulative effect of IAA in the rooting medium and IRC produced by endophytic bacteria would be complicated. Therefore, our study was mainly focused on the PGP properties of isolates that did not show significant production of IRC in vitro, including *P. frigoritolerans* S1.2, *Bacillus* sp. L3.4, and *B. toyonensis* Nt18. Only one IRC-producing strain, *B. toyonensis* S1.4, was included in the bacterial inoculum effect on in vitro propagated apple shoot rooting and ex vitro acclimatisation analysis. In addition, Gram-positive bacteria of the Bacilli class were preferred for the experiments, which are often well-adapted to survive in soil environments and colonise various plants [44].

### 2.4. Effect of Bacterial Inoculum on Apple Shoot Rooting and Acclimatisation Ex Vitro

The potential of plant growth-promoting bacteria to improve apple in vitro shoot rooting and adaptation ex vitro was assessed using four selected bacterial isolates or their combined inoculum. The first inoculation was performed during transfer to the rooting medium by submerging shoots in a bacterial suspension of density corresponding to approximately 1–5 × 10^7^ CFU mL^−1^. An additional 1 mL inoculum of bacterial suspension of the same density was applied to the root zone after planting to the substrate for ex vitro acclimatisation.

Rooting efficiency was assessed after 3 weeks of in vitro propagated shoot rooting, including one week of root initiation on a medium containing IBA and two weeks of root elongation on a medium without growth regulators. The mean rooting efficiency varied from 72 to 84% (Appendix A). Treatment with *Bacillus* sp. L3.4 and *B. toyonensis* Nt18 resulted in a noticeably larger proportion of rooted shoots (84 ± 12% and 84 ± 15%, respectively) compared to the control (73 ± 12%); however, there was considerable variation in results between experiments and between batches of shoots, and no significant difference was detected among the experimental groups.

To study ex vitro acclimatisation, rooted shoots were planted in coco fibre substrate plugs and maintained under greenhouse conditions. After one month of acclimatisation, a significant root growth stimulation was observed for experimental groups, including treatment with pure strains of *Bacillus* sp. L3.4, *B. toyonensis* Nt18, or combined inoculum, which resulted in a 53%, 36%, and 59% increase in cumulative root length, respectively (Figure 7A; Appendix A). Analysis of root number and individual root length showed similar though less pronounced results, suggesting that the increase in cumulative root length was a combined contribution of the two parameters (Appendix A). As plant leaf and other lateral organ growth is regulated by the balance between water uptake in roots and transpiration in leaves, as well as direct signals from the roots [45,46], the root growth-stimulating effect of bacterial inoculum also resulted in a significant increase in sapling leaf area and overall biomass accumulation (Figure 7B,C). Despite the PGP effect of bacterial inoculum, the survival rate after the one-month acclimatisation period remained within a margin of error among all experimental groups. Partially, this could be related to an overall high survival rate of acclimated plantlets observed for all experimental groups (87–97%). In addition, differences among the experimental groups were partially obscured by variation in data. This implies that despite the shoot culture used in the experiments originating from the single mericlone, the presence of inherent variation in the physiological and/or epigenetic status of individual shoots resulted in a heterogeneous rooting and acclimatisation response.

### 2.5. Metagenomic Analysis of Root and Root Zone Microbial Communities of Acclimatised Apple Saplings

Root samples from the control and three experimental groups that showed inoculum PGP effect during the acclimatisation were used for metataxonomic analysis to assess microbial diversity in the roots and rhizoplane of the ex vitro acclimatised apple saplings. High-throughput sequencing of the *16S rRNA* gene domain V4 yielded from 74,682 to 225,984 high-quality reads, of which 32–52% were mapped to bacterial taxa (Appendix A).

Alpha diversity analysis revealed reduced bacterial taxa diversity in all inoculum-treated experimental groups compared to the control (Figure 8; Appendix A). The effect was the most pronounced for the *B. toyonensis* Nt18 (g = 3.2 and 7.7 for the Chao1 and Shannon H indices, respectively); however, a strong effect size (g = 1.1–2.4) was also detected for the other two inoculum-treated experimental groups. The reduced bacterial richness in the inoculum-treated samples was likely related to the reduced number of low-abundance taxa (Appendix A). In addition, these samples had a 2–3 fold higher relative abundance of the predominant *Burkholderiaceae* family taxa compared to the control (an average of 30–54% and 14% of all reads, respectively).

Among the Gammaproteobacteria, the *Burkholderiaceae* family was predominant in the core microbiome of all samples, followed by *Oxalobacteraceae* (Figure 9). *Caulobacteraceae*, *Sphingomonadaceae*, and *Rhizobiaceae* families were Alphaproteobacteria prevalent in most samples. Notably, the *Bacillaceae* family, representing the inoculum species *Bacillus* sp. L3.4 and *B. toyonensis* Nt18, had a similar abundance of 0.2–0.9% of all reads, including the control. This suggests that the bacteria could not effectively colonise and propagate in the rhizosphere of acclimatised apple saplings.

A discrete presence of several taxa in control, *Bacillus* sp. L3.4, or combined inoculum-treated samples, was evident in the core microbiome data. However, due to substantial variation in microbial population composition and abundance data among the samples collected from different experiments (Appendix A), no significant difference (*p* = 0.11) in bacterial population composition was detected using PCoA. This could be related to the microbial composition differences among the separate coco fibre substrate batches used in the independent experiments, resulting in the disparity of the results.

Further, metagenomic analysis of taxonomic and functional gene diversity was used to assess the potential growth stimulation mechanism of the PGP bacteria inoculum in apple saplings during ex vitro acclimatisation. Shotgun metagenomic analysis of the root samples would be impaired by the abundance of the host plant DNA; therefore, root zone substrate samples were used for the analysis. The experimental group treated with combined inoculum, including both PGP strains, *Bacillus* sp. L3.4 and *B. toyonensis* Nt18, was compared to the control. After sequence filtering and host sequence removal, an average of 6.6 Gb per sample of high-quality sequencing data was obtained (Appendix A). The assembled metagenomes of individual samples included 802,017 to 987,908 genes, resulting in the non-redundant gene set of 2,366,576 genes with a 490 bp average length.

Contrary to sapling roots, alpha diversity analysis showed a similar richness of bacterial taxa in the root zone substrate samples. A significant increase (g = −3.5) in diversity in the inoculum-treated samples compared to the control was detected by the Simpson’s diversity index (Appendix A). This was not likely related to changes in the evenness of taxa abundance distribution, as it was rather evenly distributed in the root zone samples, with only *Opitutus* sp. marginally exceeding 3% of all reads in the inoculum-treated samples (Appendix A). Other taxa representing >0.1% of all reads at least in one of the experimental groups included genera of *Rhodoplanes* (0.5–1%), *Leifsonia* (0.15–0.5%), *Beijerinckia* (0.2%), *Tardiphaga* (0.2%), *Steroidobacter* (0.2%), *Pseudorhodoplanes* (0.1%), *Reyranella* (0.1%), *Terriglobus* (0.1–0.3%), *Flavisolibacter* (0.1–0.7%), and *Weizmannia* (0.1%) (Appendix A). As was observed in the root samples, *Bacillus* sp. abundance varied from 0.3% to 0.8% of all reads in the root zone substrate samples, but no significant difference was detected between the experimental groups. Among the predominant taxa, a significantly increased (1.2–2.6 log fold change) abundance in the inoculum-treated samples was detected only for *Leifsonia*, *Tardiphaga*, and *Reyranella* spp. Differential abundance data for these and other 32 genera of bacteria with at least a 2-fold change in abundance are shown in Figure 10.

Annotation using the Kyoto Encyclopedia of Genes and Genomes (KEGG) metabolic pathway database revealed a significant difference (*p* = 0.001) in the abundance of functional genes between the two experimental groups. Although most of the detected changes were relatively small (1.1–4.6-fold) and values of the coefficient of determination were moderate (R^2^ = 0.31–0.32), significant differences were associated with 259 KO terms and five metabolic pathways (Figure 11, Appendix A). Genes involved in photosynthetic pathways had a 2.7-fold higher abundance in the root zone of the control experimental group. Also, a slightly higher (10%) abundance of genes related to the biosynthesis of polyketide sugar units was detected in the control samples. Conversely, treatment with combined inoculum resulted in a 12–35% higher abundance of genes involved in non-ribosomal peptide and glycopeptide antibiotic biosynthesis and degradation of furfural. Metagenome data annotation using the specialised databases of microbial genes involved in nitrogen (NCycDB) and phosphorus (PCycDB) cycles did not reveal significant differences between the two experimental groups (Appendix A).

## 3. Discussion

The apple bud endophytic microbial community analysis results obtained in our study (Figure 3) were partially in line with studies that previously identified bacteria of *Sphingomonadaceae*, *Beijerinckiaceae*, *Acetobacteraceae*, *Pseudomonadaceae* families in apple buds and also in the phyllosphere of other woody and non-woody plants [33,48,49,50,51,52]. *Sphingomonas* sp., *Pseudomonas* sp., *Beijerinckiaceae* genera *1174*−*901*−*12* and *Methylobacterium,* and *Acidiphilium* spp. of *Acetobacteriaceae* were described as predominant in apple dormant buds collected from apple trees in southern Finland [33]. The *1174−901−12*, *Acidiphilium,* and two other unidentified genera of *Beijerinckiaceae*, and *Acetobacteraceae* families were also predominant in the core microbiome of phyllosphere samples collected from six distinct woody plant species in Central Europe [49]. These taxa include resilient bacteria that are ubiquitous in a variety of environments and are involved in important ecological functions often associated with PGP activity, such as mineral solubilisation, siderophore production, nitrogen fixation, or production of IRC and other phytohormones [53,54,55,56].

Another three families of the apple bud core microbiome identified in our study, including *Oxalobacteraceae*, *Sphingobacteriaceae*, and *Microbacteriaceae* (Figure 3, Appendix A)*,* were less prevalent in the apple bud samples described in the study by Roslund et al. [33], suggesting a possible regional or genotype-specific variation in the bud microbiota. Members of these families are also common in water or soil environments and have been described as plant-associated bacteria. The enrichment of *Oxalobacteraceae* family taxa in the rhizosphere is associated with PGP activity [57,58]. A diverse metabolic activity and plant growth stimulation by bacteria of *Sphingobacteriaceae* family has been described [59,60]. *Microbacteriaceae* is a large family of actinobacteria comprising 51 genera and includes several species associated with plants [61,62], one of the most prominent of which is endophytic *Curtobacterium* sp. [63]. However, the short-read data used in our study have not provided sufficient taxonomic resolution of the *Microbacteriaceae*. Therefore, a more detailed analysis would be required to establish the detailed taxonomic identity of the apple bud endophytic actinobacteria.

Contrary to dormant buds that were used for the initiation of apple in vitro shoot culture, the in vitro shoots were mainly dominated by Gammaproteobacteria of *Enterobacteriaceae* family and Bacilli such as *Bacillaceae* and *Lactobacillaceae* (Figure 3). The families include bacteria with a previously described presence in in vitro cultures [25,64,65]. This might indicate that the bacteria are adapted to survive within the plant tissues and/or cultivation medium under in vitro conditions. Plant tissues cultivated in vitro are exposed to reduced light, the presence of exogenous sugars and growth regulators, high humidity, or partial waterlogging in a semisolid medium. Such conditions disturb the development of the photosynthetic apparatus and functioning of stomata [66] and require physiological adaptations leading to changes in metabolic activity [67,68] and activation of stress response mechanisms [69]. Potentially, the changes in the balance of metabolic and/or signalling pathways could affect mutualistic interaction with endophytic bacteria.

Plant endophytic microbial communities are essential for maintaining plant health and productivity, and microbial diversity loss during in vitro cultivation could be critical for plant acclimatisation and establishment ex vitro [25,65]. Preservation of endophyte biodiversity during in vitro cultivation could be challenging. Therefore, applying PGP inoculum before acclimatisation might present a useful alternative, resulting in improved in vitro propagated germplasm performance. Among the four cultivable bacteria isolates tested in our study, an inoculum of *Bacillus* sp. L3.4 and *B. toyonensis* Nt18 stimulated root growth and biomass accumulation during apple acclimatisation ex vitro (Figure 6). As both strains, *Bacillus* sp. L3.4 and *B. toyonensis* Nt18, had no significant IRC production activity in the in vitro assay (Figure 6); likely, the PGP effect of the inoculum did not directly involve root growth stimulation by the hormone production. The inoculum bacteria *B. toyonensis* Nt18 was previously shown to stimulate the growth of in vitro tobacco shoots [26]. Over the last decade, PGP properties, including inorganic and organic phosphate solubilisation or production of siderophores, auxins or cytokinins, were demonstrated for a variety of *B. toyonensis* strains [70,71,72,73]. In addition, biocontrol potential for the bacterium was proposed based on genome analysis and in vitro assays [74,75].

Functional diversity analysis of the root zone microbiota did not indicate significant changes in the abundance of genes involved in soil nutrient metabolic cycles, such as nitrogen or phosphorus, or the production of organic acid and siderophores that could facilitate plant micronutrient uptake. Yet the increased abundance of genes associated with the biosynthesis of non-ribosomal peptides, including vancomycin-related glycopeptide compounds, might significantly contribute to microbial competition and/or pathogen control in the sapling rhizosphere (Figure 11). Non-ribosomal peptides constitute an essential part of the arsenal of the pathogen-suppressing compounds of a variety of PGP bacteria, such as *Bacillus* [76], *Pseudomonas* [77], *Burkholderia* [78] or actinobacteria [79,80] species. Notably, root samples of the bacterial inoculum-treated apple saplings had a 2–3 fold higher relative abundance of the predominant *Burkholderiaceae* family taxa compared to the control. Potentially, the pathogen-suppressing properties of the bacterium could contribute to the PGP effect of the inoculum.

Metagenomic analysis could not confirm the survival and propagation of the inoculum bacteria in the acclimatised apple plantlet roots of the rhizosphere. Notably, besides the actinobacteria of *Leifsonia* sp., acclimated apple plantlet root and root zone substrate microbiota were dominated by several taxa of Pseudomonadota phylum. A low abundance of Bacillota in the apple plantlet microbiome could be indicative either of the presence of unfavourable growth conditions (possibly a high concentration of lignocellulose degradation products such as furfural) or formation antagonistic interactions within the microbial community that would suppress the growth of the Gram-positive bacteria taxa. This could be an important limiting factor for the effective use of the inoculum treatment and would require repeated application of the inoculum treatment to improve its efficacy.

Despite the limited lifespan of the inoculum bacteria in the coco fibre substrate used for acclimatisation, it appeared to induce an enduring effect on the metabolic profile of the microbial community. Further analysis would be required to clarify whether the effect could be attributed to direct microbial-microbial interactions or could be a consequence of inoculum-induced changes in host plant physiology. Some microorganisms can affect host plants, such as host root exudations, which, in turn, could reshape the composition of the plant-associated microbiome [81]. Several changes in antimicrobial compound metabolism could contribute to reshaping the microbial community composition in the apple sapling rhizosphere. An increase in the abundance of furfural degradation genes might indicate higher lignocellulose degradation activity by wood-degrading fungi [82]. Furfural is regarded as a major toxin of microbial cells, but some bacteria, mostly limited to Gram-negative aerobic species, evolved various defence mechanisms [83]. Also, a relatively high abundance of genes involved in polyketide sugar biosynthesis in the root zone of apple saplings could be associated with the abundance of *Leifsonia* sp. actinobacteria. Polyketide compounds are found in most organisms but are especially abundant in actinobacteria and include bioactive molecules with antimicrobial activity [84]. Considering this, it could be further speculated that the change in the toxic and antimicrobial compound biosynthesis profile contributed to the reduced photosynthetic gene abundance of the inoculum-treated samples. Interestingly, phototrophic *Rhodoplanes* sp. was detected among the most prevalent bacteria in the root zone microbiome. However, its abundance was not affected by the inoculum treatment. Therefore, the changes in photosynthetic gene abundance could more likely be attributed to other photosynthetic bacteria or algae.

## 4. Materials and Methods

### 4.1. Plant Material and In Vitro Shoot Culture

Dormant bud samples of apple (*Malus* × *domestica* Borkh.) cv. Gala were collected at the end of March 2021 from two field-grown trees maintained at the collection of genetic resources of the Lithuanian Research Centre for Agriculture and Forestry, Kaunas region, Lithuania (N 55.075, E 23.809). The samples were stored on ice and processed on the same day.

To establish an in vitro shoot culture, bud explants were rinsed under running tap water and surface sterilised by submerging them in mercuric chloride solution for 10 min, followed by washing with sterile deionised water. Sterilised explants were cultured in Murashige and Skoog (MS) [85] medium supplemented with 100 mg L^−1^ myo-inositol, 1 mg L^−1^ thiamine, 0.5 mg L^−1^ 6-benzylaminopurine (BAP), 0.05 mg L^−1^ IBA, 0.1 mg L^−1^ gibberellin A3, 3% sucrose, and 0.7% plant agar (pH 5.7) at 22 ± 3 °C under illumination of 60–70 μmol m^−2^ s^−1^ intensity and a 16 h photoperiod. The established shoot culture was maintained in Steri Vent containers (107×94×96 mm) (Duchefa Biochemie B.V., Haarlem, Netherlands) and transferred to fresh medium every four weeks. For DNA analysis, in vitro shoot samples were collected during the active growth stage, 2–3 weeks after transfer to a fresh medium.

*Nicotiana tabacum* L. cv. Samsun-NN cell suspension was maintained in MS medium supplemented with 100 mg L^−1^ myo-inositol, 1 mg L^−1^ thiamine, 100 mg L^−1^ KH_2_PO_4_, 0.01 mg L^−1^ kinetin, 1.0 mg L^−1^ NAA, and 3% sucrose at 25 ± 1 °C with shaking at 80 rpm.

### 4.2. In Vitro Shoot Tip Cryopreservation Conditions

For CP treatment, in vitro shoots were cold-hardened at 4 ± 1 °C and a short-day (8 h) photoperiod of 15–20 μmol m^−2^ s^−1^ for two weeks. Apical shoot tips were excised from the shoots and subjected to a CP procedure described by Vinskienė et al. [25]. To restore shoot culture after CP, the apical shoot tips were thawed in a 40 °C water bath for 2 min, rinsed with liquid MS medium containing 1 M sucrose for 20 min., and transferred to a Petri dish with MS medium supplemented with 1 g L^−1^ polyvinylpyrrolidone and 1% plant agar. For the first week, shoot tips were incubated at 22 °C in the dark and then 60–70 μmol m^−2^ s^−1^ illumination and a 16 h photoperiod was used for 2 weeks. Regenerated shoots were maintained under in vitro shoot propagation conditions. For cryogenic freezing experiments, 2–4 replicates of 10 shoot tips were used for each experimental group.

### 4.3. In Vitro Shoot Rooting and Ex Vitro Acclimatisation

Four weeks after transfer to fresh medium, shoots were immersed in bacterial suspension in MS medium at a concentration of approximately 1–5 × 10^7^ colony-forming units (CFU) per millilitre. MS medium without bacteria was used for the control treatment. After several minutes of incubation, shoots were transferred to Steri Vent containers with rooting initiation medium (RIM) containing 0.5× MS medium supplemented with 100 mg L^−1^ myo-inositol, 1 mg L^−1^ thiamine, 0.3 L^−1^ IBA, 1.5% sucrose, and 0.7% plant agar (pH 5.7). Twelve to fourteen shoots were used in each container. After one week of incubation at 25 °C in the dark, shoots were transplanted to a fresh root elongation medium with the same composition as RIM, except without IBA, and maintained under the same conditions used for shoot propagation for two weeks. The experiment was repeated 4–5 times.

Coco coir fibre growing plugs (∅ 3 cm, Kesko Senukai Lithuania, Kaunas, Lithuania) were used for ex vitro shoot acclimatisation. Plugs (up to 35 pcs) were placed into a tray (28 × 28 × 10 cm) and soaked in deionised water supplemented with 1:1000 diluted COMPO Complete liquid fertiliser concentrate (COMPO GmbH, Münster, Germany). Rooted shoots were transferred to the substrate plugs and inoculated with 1 mL of endophytic bacteria suspension at approximately 1–5 × 10^7^ CFU mL^−1^. Deionised water without bacteria was used for the control treatment. Plantlets were maintained at room temperature under 100–150 μmol m^−2^ s^−1^ intensity illumination and a 16 h photoperiod. During the first two weeks of acclimatisation, plantlets were covered with a transparent plastic cover to maintain high humidity. After four weeks of acclimatisation, sapling survival, biomass accumulation, root length, and leaf area were assessed, and root and root zone samples were collected for DNA-based microbial population composition and function analysis.

### 4.4. Redox-Modulating Activity Screening Using Tobacco Cell Culture

Co-cultivation of tobacco cell suspension and bacterial isolate was carried out according to the protocol described by Tamošiūnė et al. [24], and redox activity was assessed using 2’,7’-dichlorodihydrofluorescein diacetate (H_2_DCF-DA) staining, according to Joo et al. [86]. Briefly, approximately 100 mg fresh weight of tobacco cell suspension was resuspended in 2 mL of fresh medium, and the bacteria were added at a final density of 0.1 OD_600_ after being centrifuged and resuspended in MS medium. After incubation on the Wisd Digital Rotator RT-10 (Witeg Labortechnik GmbH, Wertheim, Germany) for 6 h, cells were aliquoted into a 96-well plate, and H_2_DCF-DA was added at a final concentration of 10 µM and incubated for 30 min at room temperature. Cell fluorescence was assessed using a fluorometer LS55 (Perkin-Elmer, Waltham, MA, USA) with Ex = 485 nm and Em = 525 nm wavelength. To correct for cell quantity variation among the wells, cell autofluorescence was measured at Ex = 350 nm and Em = 440 nm wavelength before staining with H_2_DCF-DA and used for data normalisation within one experimental group.

### 4.5. Tobacco Seedling Growth-Modulating Activity of Bacterial Isolates

Seedling co-cultivation with bacterial isolates was conducted according to Tamošiūnė et al. [40]. *N. tabacum* L. cv. Samsun-NN seeds were sterilised with 5% sodium hypochlorite for 5 min, followed by 70% ethanol, washed 5 times with sterile deionised water, and stored at 4 °C for 48 h. Bacterial isolates cultured in liquid lysogeny broth (LB) [87] overnight were centrifuged and resuspended in deionised water to approximately 1–5 × 10^7^ CFU mL^−1^. Bacterial suspension was applied to tobacco seeds, and 1 mL aliquots of suspension were spread on a sterile filter paper in a Petri dish. Deionised water without bacteria was used for the control treatment. After inoculation, the plates with seeds were incubated at 22 ± 3 °C, under 16 h photoperiod illumination of 150 µmol m**^−^**^2^ s**^−^**^1^ intensity for 6 days. To assess root growth, seedlings were stained with 1 mg mL^−1^ nitro blue tetrazolium dissolved in 50 mM NaPO_4_, 0.02% NaN_3_, and fixed with ethanol/glycerol/acetic acid (3:1:1). Images were captured using a Nikon SMZ1000 microscope (Nikon, Tokyo, Japan). The seedling radicle and hypocotyl length were measured using ImageJ software v. 1.54 [88].

### 4.6. DNA Sample Preparation

The dormant bud samples were sterilised, as described by Hata et al. [89]. Briefly, buds were incubated in 70% ethanol for 5 min, 15% hydrogen peroxide for 20 min, and 70% ethanol for 1 min and washed with sterile distilled water five times. Then, the upper layer of the scales was peeled off with a scalpel. To confirm sterility, 100 µL of the last rinsed water was inoculated on LB plates and incubated at room temperature for three days. Sterilisation was considered successful when no colonies were observed. Pooled samples of 5–7 bud cores were used for DNA preparation. In vitro propagated and CP-treated shoot samples were pooled from 25 to 30 shoots collected in the middle of the propagation period (two weeks after transfer to fresh medium). Sapling root and root zone substrate samples were collected after four weeks of acclimatisation. Saplings were uprooted, roots were washed with deionised water and dried with a towel, and pooled samples were prepared from 15 to 30 sapling roots. Root zone substrate samples pooled from 12 to 20 saplings were frozen at −70°C, lyophilised, and stored at −20 °C until use.

For DNA extraction, sterilised bud, in vitro propagated shoot, CP-treated shoot, and sapling root samples were flash-frozen in liquid nitrogen and ground to a fine powder. From the plant material and lyophilised root zone substrate samples, DNA was prepared using the cetyltrimethylammonium bromide extraction-based method described by Ding et al. [90], except 0.2 g of starting material was used, and the final elution was performed in 20–30 µL of nuclease-free water. DNA extracted from the root zone substrate was additionally purified using MagMax Pure Bind beads (Applied Biosystems, Waltham, MA, USA) according to manufacturer’s instructions.

### 4.7. DNA Library Preparation and Sequencing

For dormant bud samples, six replicates collected from two trees (samples B1–3 and B4–6, respectively) located in different parts of the collection, and three replicates for each experimental group of the in vitro propagated (IV1–3) and CP-treated shoots (CP1–3) were used in the metataxonomic analysis. A fragment of variable domain 4 (V4) of the *16S rRNA* gene was amplified using primers (Uni516F 5′-CCAGCAGCCGCGGTAATA-3′ [91] and V4R 5′-GGACTACCAGGGTATCTAATCCTGT-3′ [92]), and DNA library for Ion Torrent sequencing using the Ion PGM system (Thermo-Fisher Scientific, Waltham, MA, USA) was prepared as described previously [35]. Base calling and run demultiplexing were performed by Torrent Suite v.5.15 (Thermo-Fisher Scientific, USA) with default parameters.

For sapling root metataxonomic analysis, four replicates were used for each experimental group (control (C1–3), *Bacillus* sp. L3.4 (L1–3), *B. toyonensis* Nt18 (N1–3), and combined inoculum (CI1–3)). Fragment of domain V4 was amplified using primers 515F 5′-GTGYCAGCMGCCGCGGTAA-3′ [93] and 806R 5′-GGACTACNVGGGTWTCTAAT-3′ [94]. Amplicon DNA fragment library preparation and sequencing were performed at BMKGENE (Münster, Germany) using the Illumina NovaSeq system (Illumina, San Diego, CA, USA).

Metataxonomic analysis sequence data were processed using R v.4.2.1 [95]. Primer sequences were removed using Cutadapt v.5.0 [96]. Read filtering and trimming were performed with the *DADA2* v.1.8 [97] pipeline with the following parameters for unpaired reads: truncLen = c(220), maxN = 0, maxEE = c(2), truncQ = 2; for pair-end reads, the parameters were as follows: truncLen = c(220, 210), maxN = 0, maxEE = c(2, 2), truncQ = 2. After merging the pair-end reads and chimaera removal, taxonomy was assigned using the *DADA2* naïve Bayesian classifier and Silva v.138.2 database [98].

For short-read shotgun metagenomic analysis of microbial taxonomic and functional diversity in the sapling root zone substrate, control (C1–3) and combined inoculum (CI1–3) experimental groups (three replicates each) were used. Short-read DNA fragment library preparation and sequencing were performed at BMKGENE (Münster, Germany) using paired-end chemistry (PE150) on the Illumina NovaSeq system (Illumina, San Diego, CA, USA). After filtering with fastp v.0.24.0 [99] with a quality cutoff of 20% and reads shorter than 30 bp, host genomic DNA was removed using Bowtie2 v.2.5.4 [100] with *M. domestica* cv. Gala genome [101] sequences as reference. Contigs shorter than 300 bp were removed, and metagenomes were assembled using MEGAHIT v.1.2.9 [102] and evaluated using QUAST v.3.2 [103]. MetaGeneMark v.3.26 [104] was used for gene prediction. MMseqs2 v12-113e3 [105] was used to remove redundancy with a protein sequence similarity threshold of 90% and a coverage threshold of 80%. Protein sequences were annotated based on the KEGG release 111.1 [106], NCycDB released in 2019 [107], and PCycDB v.1.1 [108] databases with an e-value ≤ 1 × 10^−5^ using Diamond v2.1.11 [109]. Non-redundant genes were aligned against NCBI nr database (released on 15 August 2024) [110] to obtain taxonomy annotation.

### 4.8. Data Analysis

Data distribution normality was assessed by the Shapiro–Wilk test using the R package *dplyr* v.1.1.4 [111]. Statistically significant differences between the means of the two experimental groups were assessed by Student’s *t*-test using the R package stats v.4.5.0 [95]. Statistically significant differences between the means of three or more experimental groups were assessed using the R package *agricolae* v.1.3-7 [112], and ANOVA analysis and Tukey’s post hoc test or Kruskal–Wallis test and Fisher’s least significant difference post hoc test were used for normal or non-normal distribution data, respectively. Data were visualised by the *ggplot2* v.3.5.1 package [113]. The effect size of significant differences was estimated and visualised using the R package *Durga* v.2.0 [114].

A taxonomy-based dendrogram of bacterial isolates was built by the hierarchical cluster analysis using the daisy function and Gower distance matrix of the R package *cluster* v.2.1.6 [115].

Statistical data analysis of *16S rRNA*-based metataxonomic data was performed using the Microbiome Analyst server v.2.0 [36]. Data were rarefied to the minimum library size, and total sum normalisation was applied to taxonomic count data by dividing feature read counts by the total number of reads in each sample. ASVs with <10 counts and <10% prevalence in the sample were removed, data were rarefied to the minimum library size, and total sum normalisation was applied by dividing feature read counts by the total number of reads in each sample. Observed, Chao1, ACE, Fisher’s alpha, Shannon (H), and Simson’s (D) indexes were used to estimate bacterial species richness and evenness within samples. PCoA and Bray–Curtis dissimilarity metric were used to compare the diversity of bacterial communities among the experimental groups. The threshold of >5% counts and >30% samples were used for the core microbiome calculation.

Short-read shotgun metagenomic data were normalised by cumulative sum scaling of gene counts using the *cumNorm* function in the *metagenomeSeq* R package v.1.22.0 [47]. A zero-inflated Gaussian model was fitted for each gene using the function *fitZig*, and *p*-values were corrected using the BH FDR method implemented in the *MRfulltable* function in *metagenomeSeq*. Based on the distance matrices obtained by beta diversity analysis, a Permutational Multivariate Analysis of Variance (PerMANOVA) with a binary Jaccard distance matrix was performed using the R package *vegan* v.2.6-10 [116].

## 5. Conclusions

Our study provided new insights into the structure of the endophytic bacteria community of the field-grown tree dormant buds of apple cv. Gala and its changes induced by in vitro cultivation and CP treatment. Bacterial taxa previously described as endophytes of apple and other plants were predominant in dormant buds, whereas the microbial population of in vitro shoot culture was dominated by bacteria potentially better adapted to the in vitro environment. The findings contribute to an understanding of the role of plant-associated microbiota in producing horticultural crop propagation material using in vitro techniques.

This study showed that PGP bacteria could improve the performance of the in vitro propagated germplasm. The inoculum of *Bacillus* sp. L3.4 or/and *B. toyonensis* Nt18 stimulated the growth vigour of apple plantlets during ex vitro acclimatisation. The plantlet growth stimulation was likely related to the pathogen-suppressing and/or plant stress-mitigating effect. However, further research is necessary to enhance bacterial inoculum colonisation in the treated plant rhizosphere. This could be achieved by selecting strains better suited to the host and growth environment and designing microbial consortia that facilitate the propagation of the inoculated bacteria. In addition, investigating the physiological mechanisms underlying the apple shoot and plantlet response to the endophytic bacteria inoculum treatment would be an important area of research for future studies.

## Figures and Tables

**Figure 1 plants-14-01045-f001:**
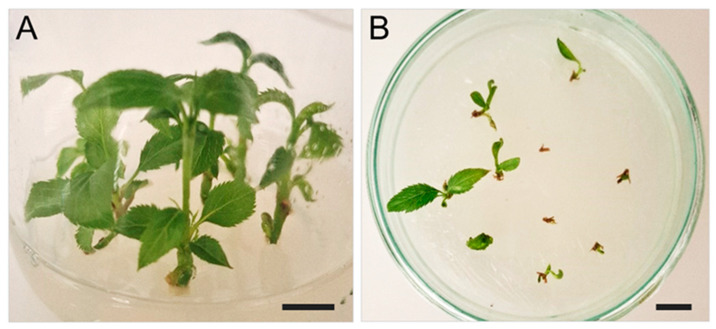
Representative images of apple cv. Gala in vitro shoot culture (**A**) and shoots regenerated from CP-treated apical shoot tips (**B**). The scale bar represents 1 cm.

**Figure 2 plants-14-01045-f002:**
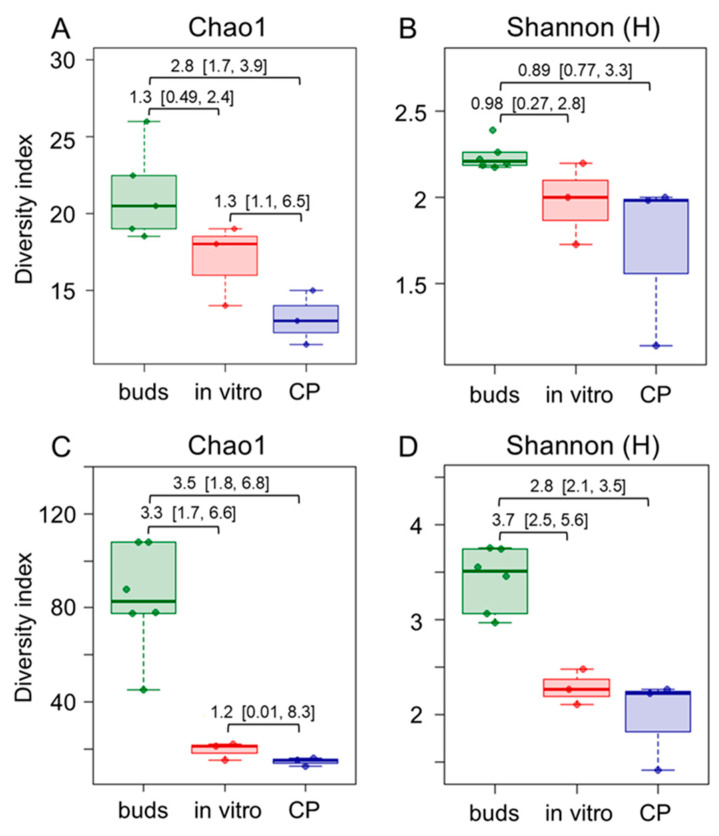
Comparison of family- (**A**,**B**) and ASV feature-level (**C**,**D**) endophytic bacteria taxonomic richness (**A**,**C**) and diversity (**B**,**D**) in apple dormant bud (buds), in vitro shoot (in vitro), and cryopreservation-treated in vitro shoot (CP) tissue samples. Alpha diversity indices were estimated using *16S rRNA* domain V4 amplicon sequencing data. The data are shown as boxplots representing the medians, minimum and maximum scores, and lower and upper quartiles; numbers denote Hedge’s g standardised mean difference effect size with a 95% confidence interval shown in the brackets.

**Figure 3 plants-14-01045-f003:**
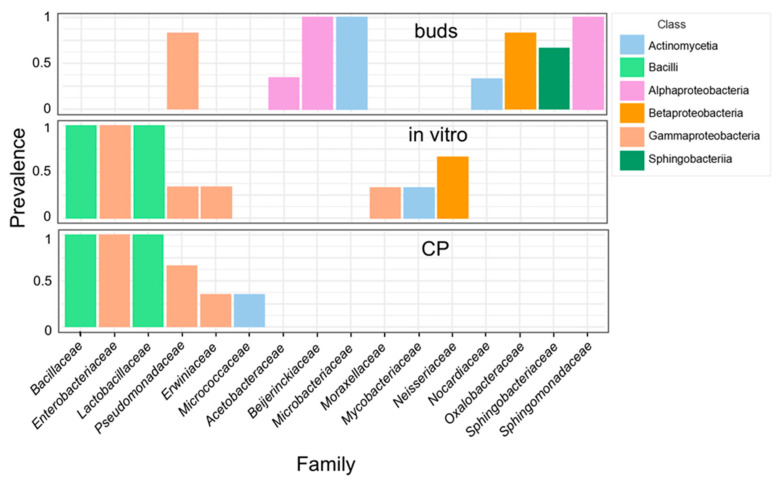
Core microbiome of apple dormant bud (buds), in vitro shoot (in vitro), and cryopreservation-treated (CP) in vitro shoot tissues. Family-level taxonomic compositions were estimated using *16S rRNA* variable region V4 amplicon metataxonomic sequencing. The threshold of >5% counts and >20% samples was calculated using the MicrobiomeAnalyst server [36]. Different colours represent bacterial classes.

**Figure 4 plants-14-01045-f004:**
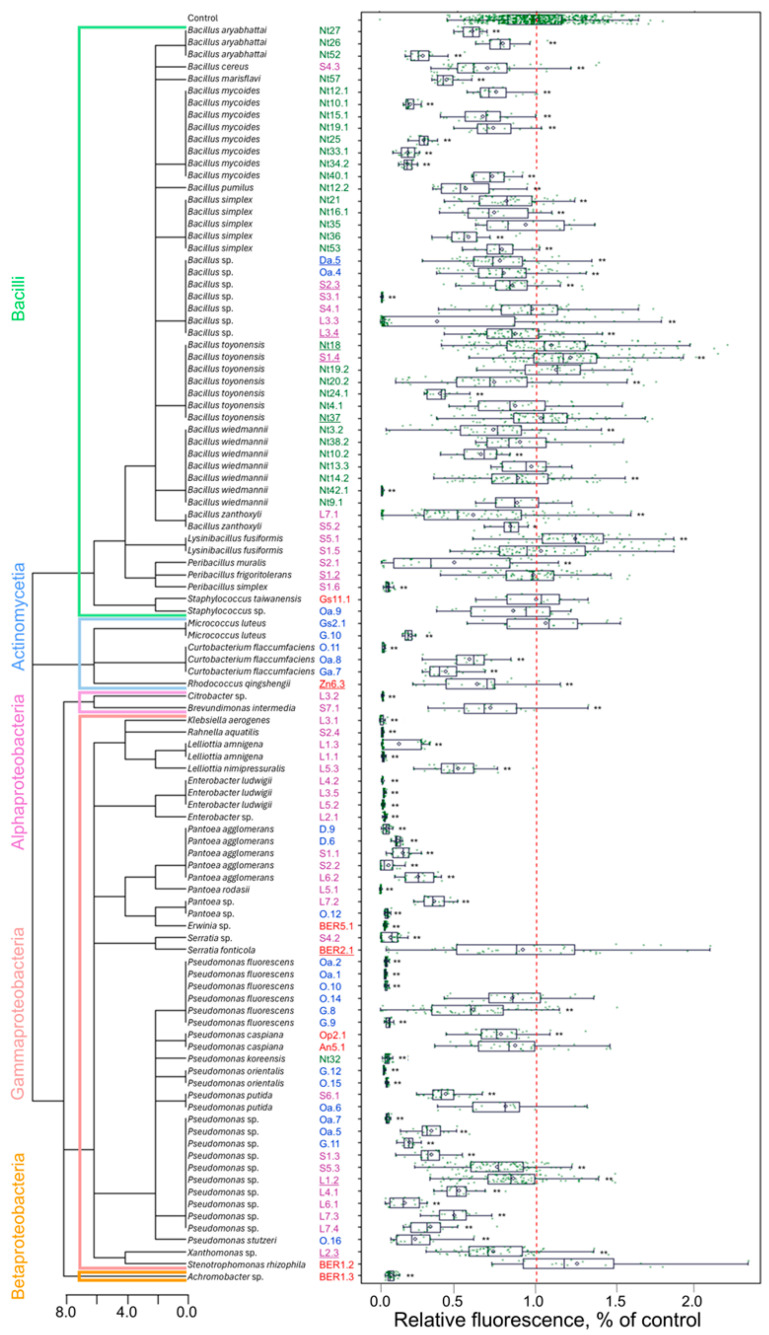
Tobacco cell redox-modulating activity of endophytic bacteria isolates. The isolate identification numbers shown in blue, green, red, and magenta font indicate isolates obtained from apple, tobacco, strawberry, or *R. rosea* plants, respectively. The underlined numbers denote isolates used for analysing tobacco plant growth regulation in vitro. The tree diagram represents the taxonomic relationship among the isolates. The redox activity of tobacco cells was assessed based on 2′,7′-dichlorodihydrofluorescein diacetate (H_2_DCF-DA) fluorescence, and the relative fluorescence data (as % of control) are shown as boxplots representing the medians, minimum and maximum scores, and lower and upper quartiles; data points are plotted as green dots; the red dashed line indicates the mean value of control; asterisks * and ** denote a significant mean difference compared to control at *p* < 0.05 and *p* < 0.01, respectively.

**Figure 5 plants-14-01045-f005:**
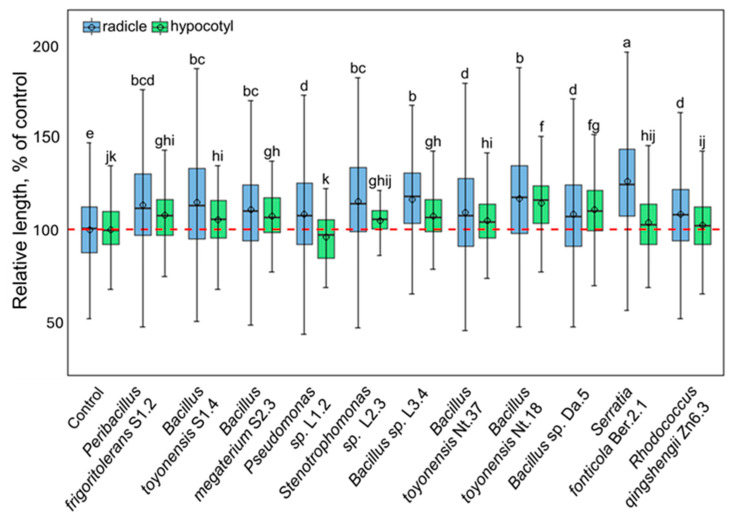
The growth-modulating effect of selected endophytic bacterial isolates on the tobacco seedling radicle (blue fill colour) and hypocotyl (green fill colour). The data are shown as boxplots representing the mean, median, minimum, and maximum scores and lower and upper quartiles; the red dashed line indicates the mean value of control; different letters denote significant differences between the analysed groups (*p* ≤ 0.05).

**Figure 6 plants-14-01045-f006:**
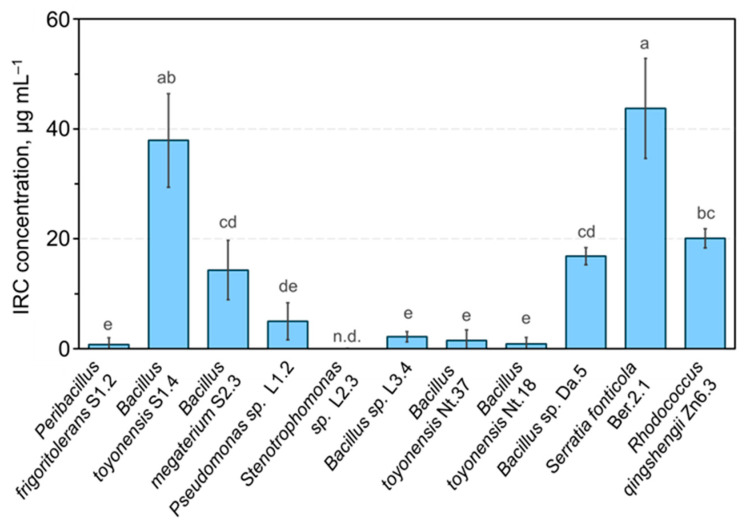
Indole-related compound-producing activity of endophytic bacteria isolates. Data are presented as the mean and standard error of the mean; different letters denote significant differences between the analysed groups (*p* ≤ 0.05). Abbreviations: IRC—indolyl-related compound; n.d.—not detected.

**Figure 7 plants-14-01045-f007:**
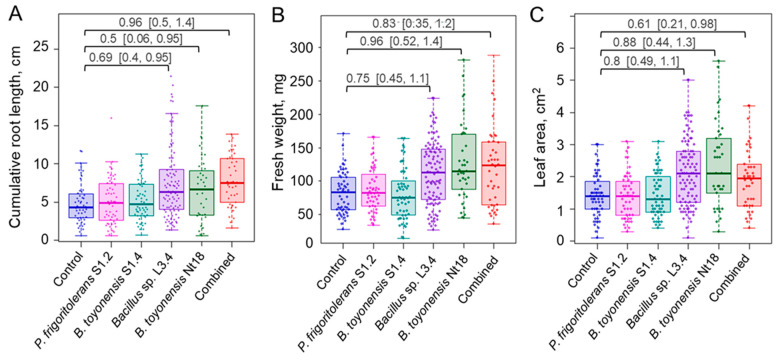
Endophytic bacteria inoculum effect on cumulative root length (**A**), leaf area (**B**), and biomass accumulation (**C**) of ex vitro acclimatised apple plantlets. The data are shown as boxplots representing the medians, minimum and maximum scores and lower and upper quartiles; data points are plotted as dots; numbers denote Cohen’s d standardised mean difference effect size with a 95% confidence interval shown in the brackets.

**Figure 8 plants-14-01045-f008:**
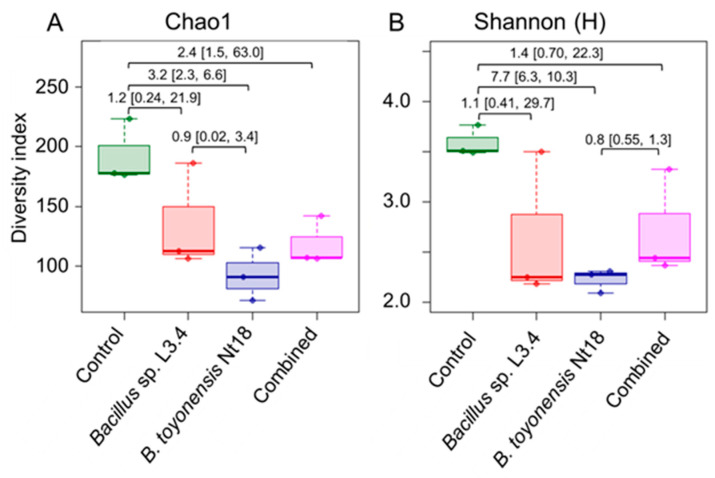
Comparison of family-level bacteria taxonomic richness (**A**) and diversity (**B**) in control and bacterial inoculum-treated ex vitro acclimated apple sapling root samples. Alpha diversity indices were estimated using *16S rRNA* domain V4 amplicon sequencing data. The data are shown as boxplots representing the medians, minimum and maximum scores, and lower and upper quartiles; numbers denote Hedge’s g standardised mean difference effect size with a 95% confidence interval shown in the brackets.

**Figure 9 plants-14-01045-f009:**
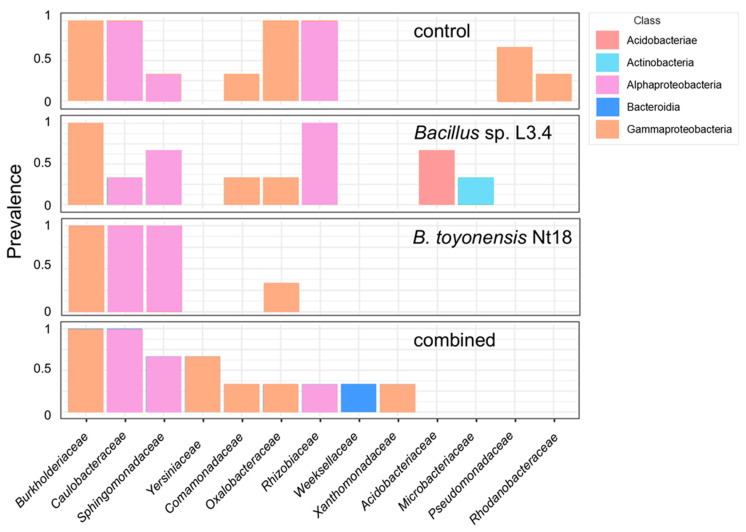
Core microbiome of control and bacterial inoculum-treated ex vitro acclimated apple sapling roots. Family-level taxonomic compositions were estimated using *16S rRNA* variable region V4 amplicon metataxonomic sequencing. The threshold of >5% counts and >20% samples was calculated using the MicrobiomeAnalyst server [36]. Different colours represent bacterial classes.

**Figure 10 plants-14-01045-f010:**
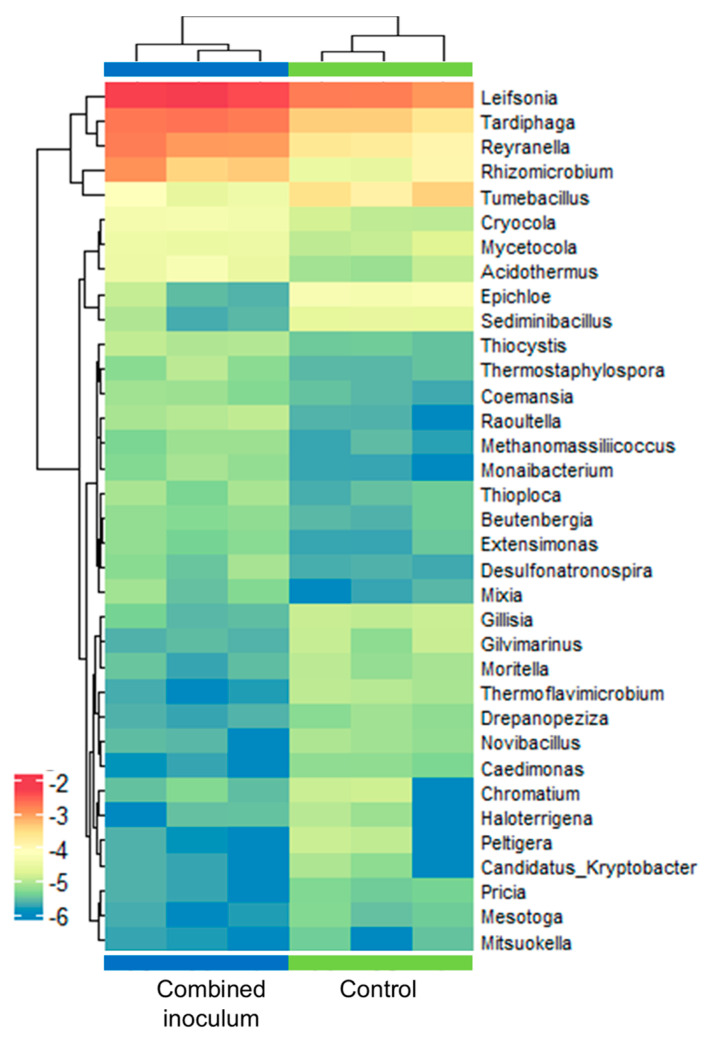
Genus-level differential abundance of bacteria in control and combined inoculum-treated root zone samples of ex vitro acclimatised apple plantlets. The heat map represents the log of normalised relative abundance data with logFC > 1 and adjusted *p* < 0.05 as defined using the *metagenomeSeq* package v.1.22.0 [47]. Each row stands for a genus, and each column stands for a sample.

**Figure 11 plants-14-01045-f011:**
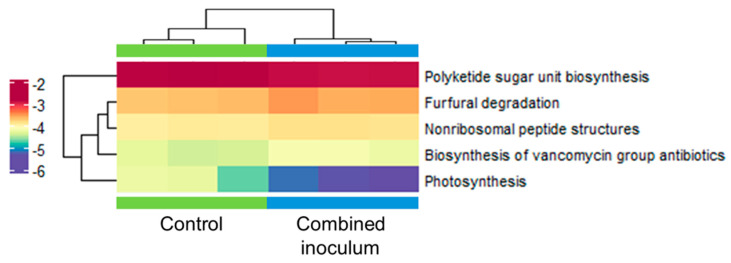
Abundance heat map of metabolic pathways, including differential abundance genes of the control and combined inoculum treated root zone substrate samples of ex vitro acclimatised apple plantlets identified using the KEGG database with *p* < 0.05 defined using the *metagenomeSeq* package v.1.22.0 [47]. KEGG pathway names are shown on the right. Each row stands for a gene, and each column stands for a sample. The colour indicates normalised relative abundance.

## Data Availability

The original contributions presented in this study are included in the article/Appendix A. Further inquiries can be directed to the corresponding author.

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
