# Peer review of "Inoculum of Endophytic *Bacillus* spp. Stimulates Growth of Ex Vitro Acclimatised Apple Plantlets"

_plants, 2025, doi:10.3390/plants14071045_

Round 1

Reviewer 1 Report

Comments and Suggestions for Authors

Inoculum of endophytic Bacillus spp. stimulates growth of ex vitro acclimatised apple plantlets. The manuscript by Vinskienė et al. systematically investigated the effects of in vitro cultivation and cryopreservation (CP) on the endophytic microbiota of apple tissues using metataxonomic analysis and bacterial strain screening. It further validated the potential of plant growth-promoting (PGP) bacterial inoculum to enhance the ex vitro acclimatization of propagated apple shoots. The research revealed differences in endophytic bacterial communities between field-grown dormant buds and in vitro-propagated or CP-treated shoots, highlighting dynamic shifts in the microbiome during in vitro cultivation. Notably, Bacillus sp. L3.4, B. toyonensis Nt18 and combined inoculum with growth-promoting effects were identified, offering promising tools for optimizing in vitro propagation protocols. While the study demonstrates innovation, certain methodological and interpretive aspects require refinement to strengthen the rigor of the data and the reliability of the conclusions.

1 Results: 3.3

The authors did not explain the criteria or rationale for selecting endophytic bacterial isolates from apple (23 isolates), tobacco (32 isolates), Rhodiola (40 isolates), and strawberry tissues. It remains unclear why these specific plant species and tissue sources were chosen. A clarification is needed regarding the scientific basis for selecting these particular plant-endophyte combinations and their relevance to the study’s objectives.

2 Results: 3.3

The authors assessed the potential of bacterial isolates to interact with plant cells by co-cultivating them with tobacco cell suspension cultures and measuring reactive oxygen species (ROS) production suppression. However, relying solely on ROS inhibition as an indicator of plant-microbe interaction may be insufficient. Additional physiological indicators (e.g., cell viability assays, antioxidant enzyme activity, or gene expression analysis) should be included to strengthen the reliability of the conclusions.

3 Figure 6:

Although several isolates showed high indole-related compound (IRC) production, only B. toyonensis S1.4 (with significant IRC synthesis in vitro) was selected for further experiments. The authors attributed this to potential interference from exogenous IAA in the rooting medium. To validate the IRC data, it is recommended to design experiments using IAA-free media to eliminate confounding effects and confirm the role of bacterial-derived IRC in root growth stimulation.

4 Results:3.4 The significant promotion of leaf area and biomass accumulation by bacterial inoculum warrants further investigation into underlying physiological mechanisms. Additional measurements, such as chlorophyll content, photosynthetic efficiency, or carbohydrate metabolism analysis, could help link microbial effects to photosynthetic performance and provide a more comprehensive understanding of growth stimulation.

5 Some experimental groups (e.g., CP-treated shoots) had limited biological replicates, and significant data variability was observed. Increasing the number of biological replicates would enhance statistical power and improve the consistency and reliability of the results.

Author Response

We thank the Reviewer for taking the time to review the manuscript and for insightful comments and suggestions. Please find the detailed responses below and the corresponding revisions/corrections in track changes in the re-submitted files.

Comment 1:

Results: 3.3. The authors did not explain the criteria or rationale for selecting endophytic bacterial isolates from apple (23 isolates), tobacco (32 isolates), Rhodiola (40 isolates), and strawberry tissues. It remains unclear why these specific plant species and tissue sources were chosen. A clarification is needed regarding the scientific basis for selecting these particular plant-endophyte combinations and their relevance to the study’s objectives.

Response 1:

A collection of 104 isolates was screened to select PGP bacteria for the inoculation experiments. These included 23 isolates from apple cultivars, which represented the same species as used in the inoculation experiments. However, several studies reported the growth-promoting effects of endophytes on non-host plants; therefore, endophytic bacteria isolates obtained from plant species such as tobacco, Rhodiola, and strawberry were also included in the study. These bacteria were previously isolated at our laboratory, and potential PGP properties of the isolates were previously characterised using biochemical and genetic tests or plant in vitro assays as described in references provided in the text. The Results section was modified to explain the rationale of the isolate selection (lines 189-197).

Comment 2:

Results: 3.3

The authors assessed the potential of bacterial isolates to interact with plant cells by co-cultivating them with tobacco cell suspension cultures and measuring reactive oxygen species (ROS) production suppression. However, relying solely on ROS inhibition as an indicator of plant-microbe interaction may be insufficient. Additional physiological indicators (e.g., cell viability assays, antioxidant enzyme activity, or gene expression analysis) should be included to strengthen the reliability of the conclusions.

Response 2:

Production of reactive oxygen species is one of the earliest cellular responses following recognition of microbe-associated molecular patterns; therefore, an analysis of the intracellular redox balance of plant cells could provide insights into the capacity of the microorganisms to engage in endophytic interaction with plant cells. Therefore, this approach was used as the first step in selecting candidates for further analysis of PGP properties. Further, germinating tobacco seedling growth-modulating effects were assayed using the selected 11 isolates. In addition, the indole-related compound-producing activity of the selected endophytic bacteria isolates was assessed using an in vitro assay. To clarify the details of the isolate screening procedure, the Results section was modified (lines 199-206).

Comment 3:

Figure 6: Although several isolates showed high indole-related compound (IRC) production, only B. toyonensis S1.4 (with significant IRC synthesis in vitro) was selected for further experiments. The authors attributed this to potential interference from exogenous IAA in the rooting medium. To validate the IRC data, it is recommended to design experiments using IAA-free media to eliminate confounding effects and confirm the role of bacterial-derived IRC in root growth stimulation.

Response 3:

Paragraph (lines 232-240) of the Results section was modified to clarify the interpretation of the data presented in Fig. 6. IAA was not used in the medium for the tobacco seedling growth stimulation analysis. The concern about the cumulative effect of IAA in the rooting medium and IRC produced by endophytic bacteria was related to the in vitro propagated apple shoot rooting experiments. Since IAA was used in the root-induction medium, interpretation of the cumulative effect of IAA included in the rooting medium and IRC produced by endophytic bacteria would be complicated. Therefore, our study mainly focused on the PGP properties of isolates that did not show significant production of IRC in vitro.

Comment 4:

Results:3.4

The significant promotion of leaf area and biomass accumulation by bacterial inoculum warrants further investigation into underlying physiological mechanisms. Additional measurements, such as chlorophyll content, photosynthetic efficiency, or carbohydrate metabolism analysis, could help link microbial effects to photosynthetic performance and provide a more comprehensive understanding of growth stimulation.

Response 4:

Thank you for pointing this out. To keep the article concise, our study mainly focuses on bacterial community taxonomic structure and functional diversity dynamics. However, we agree that investigating physiological mechanisms underlying apple shoot and plantlet response to the endophytic bacteria inoculum treatment would be an important area of research for future studies. The statement about future perspectives of the study was included in the Conclusions section (lines 717-720).

Comment 5:

Some experimental groups (e.g., CP-treated shoots) had limited biological replicates, and significant data variability was observed. Increasing the number of biological replicates would enhance statistical power and improve the consistency and reliability of the results.

Response 5:

The Reviewer is correct in pointing out a significant variation in the Shannon diversity index for CP-treated shoot samples in Fig. 2. However, in other respects, the metataxonomic data of bud, in vitro and CP-treated samples were relatively consistent (CV was less than 10% for most of the data), and it was sufficient to reveal significant differences among the experimental groups. In the manuscript, we expressed some concern (lines 280-282 and 298-302) about a variation of data in the in vitro propagated shoot rooting and ex vitro acclimatisation experiments where results could be affected by different properties of shoot and substrate batches used in the experiments. However, 20-30 shoots were used in each experimental group in these experiments, and the experiments were repeated 4–5 times. Therefore, we consider the experiment design appropriate for revealing prominent changes induced by bacterial inoculum.

Reviewer 2 Report

Comments and Suggestions for Authors

Research studies in applied biotechnologies are largely focused on investigating the potential of endophytes in promoting growth and physio-chemical attributes in plants. In addition, the co-inoculation of compatible endophytic strains has also been reported to boost crop yield and productivity.

In the study, the authors investigated the changes in the diversity of endophytic bacteria and its potential to improve plant growth in apple tissues. Furthermore, a variation in bacterial diversity was observed on metataxonomic analysis, some families were more prevalent than others. What is the outcome of alterations in endophytic bacterial communities on plant growth and survival? Discuss.

This study provides important information and validates how endophytic bacteria could help promote ex vitro acclimatization of apple plantlets.

What are the major challenges in the field applications of endophytic bacteria or bacterial consortia? What are the steps taken to address it?

With increased research on endophytes' promotion of plant growth, how can the present study contribute to advancement in this field? Explain.

Most of the cited literature in the paper is relatively old; please consider including research studies carried out within 5 years.

Comments on the Quality of English Language

The manuscript required moderate english revisions for clarity.

Author Response

We thank the Reviewer for taking the time to review the manuscript and for insightful comments and suggestions. Please find the detailed responses below and the corresponding revisions/corrections in track changes in the re-submitted files.

Comment 1:

In the study, the authors investigated the changes in the diversity of endophytic bacteria and its potential to improve plant growth in apple tissues. Furthermore, a variation in bacterial diversity was observed on metataxonomic analysis, some families were more prevalent than others. What is the outcome of alterations in endophytic bacterial communities on plant growth and survival? Discuss.

Response 1:

Metataxonomic analysis revealed a significant difference in endophytic bacteria diversity in the in vitro propagated apple shoots compared to dormant bud tissues used as a source for the in vitro culture. Whereas the dormant buds included a variety of endophytic bacteria previously found in apple and other plant tissues, in vitro cultures were dominated by bacteria that were likely better adapted to the in vitro environment and were previously observed in plant in vitro cultures. A detailed description of these differences is provided in the first three paragraphs of the discussion (lines 410-448). There, we conclude that the loss of microbial diversity during in vitro cultivation could be critical for plant growth and acclimatisation since plant endophytic microbial communities play an essential role in maintaining plant health. However, using family-level data from the metataxonomic analysis, it would be difficult to pinpoint a precise contribution of specific taxa; therefore, we avoided excessive speculations on this aspect in the discussion, and detailed studies would be required to address this question.

On the other hand, analysis of the bacterial inoculum-treated apple plantlet root samples revealed a 2–3 fold higher relative abundance of the predominant Burkholderiaceae family than control. This was corroborated by the functional diversity data, which showed an increased abundance of genes involved in the biosynthesis of non-ribosomal peptides in the inoculum-treated root zone substrate samples. Since bacteria of the Burkholderiaceae were previously shown to produce non-ribosomal peptides involved in pathogen suppression, it could be proposed that pathogen-suppressing properties of the bacterium could contribute to the PGP effect of the inoculum (lines 473-477).

Comments 2:

This study provides important information and validates how endophytic bacteria could help promote ex vitro acclimatization of apple plantlets. What are the major challenges in the field applications of endophytic bacteria or bacterial consortia? What are the steps taken to address it?

Response 2:

Several challenges limit the effectiveness of endophytic bacteria application under field conditions. Although our study was not specifically designed to address these issues, and they were not directly addressed in the discussion, several aspects could be relevant to this question.

One of the important factors is bioinoculant stability and viability. In our study, for the inoculum experiments, we selected Gram-positive bacteria of the Bacilli class that are known to produce durable endospores and are well-adapted to survive in soil environments and colonise various plants. However, the study revealed that the viability of the inoculum could not be maintained in the substrate for an extended period. This led to the conclusion that more research is required to improve bacterial inoculum colonisation of the treated plant rhizosphere. This could be done by selecting more host- and growth-environment-adapted strains and modelling microbial consortia, which support the propagation of the inoculum bacteria. This problem is discussed in the manuscript (lines 478-487; 706-707; 713-717), and the text was further modified to clarify this issue (lines 240-242). 

Another important factor for effectively applying endophytic bacteria is microbial competition in the rhizosphere. This aspect often suffers from an inadequate understanding of plant–microbe-microbe interactions in the soil environments. Therefore, to adequately address this issue, we included extensive DNA-sequencing-based analysis that provided essential insights into the inoculum effect on the plant root microbial community as well as taxonomic and functional diversity in the root zone of the acclimatised apple plantlets, as discussed in the last paragraph of the Discussion section.

Comments 3:

With increased research on endophytes' promotion of plant growth, how can the present study contribute to advancement in this field? Explain.

Response 3:

There are two main aspects to which the study outcomes would contribute to advancement in this field of endophytic bacteria application. First, the study provided new insights about the effect of in vitro cultivation techniques on endophytic bacteria diversity in plants, which would contribute to the understanding of the role of plant-associated microbiota in ex-situ preservation of genetic resources and production of plant propagation material of clonally propagated horticultural crops. The Conclusions section was modified to clarify the issue (lines 695-697).

Secondly, the study results showed that applying PGP bacteria can potentially improve the performance of the in vitro propagated germplasm and reflect on associated challenges. This question is addressed in the Conclusions section, lines 713-722.

Comments 4:

Most of the cited literature in the paper is relatively old; please consider including research studies carried out within 5 years.

Response 4:

Approximately 30% of the cited literature was published within 5 years, and most references in the discussion were published in the last decade. However, several historical aspects of the development of in vitro techniques, which were reviewed in the introduction, required citing original sources published more than a decade ago. Also, several plant cultivation and microbial techniques or even metagenomic DNA sequence analysis methods described in the Methods section were published more than 5 years ago. However, to address the reviewer’s suggestion, we included several additional references to improve the representation of the previously published research relevant to the study (references 22, 41, 43).

Comments on the Quality of English Language

Comment 1:

The manuscript required moderate English revisions for clarity.

Response 1:

The manuscript text was reviewed, and changes were made to improve its clarity (changes indicated with Markup).

Round 2

Reviewer 1 Report

Comments and Suggestions for Authors

The conclusion section is overly lengthy. It is suggested to present the findings in a more concise and focused manner, avoiding extraneous details unrelated to the core conclusions.

Author Response

We thank the Reviewer for suggestions how to improve manuscript. Please find corresponding revisions in the re-submitted files.

Comment:

The conclusion section is overly lengthy. It is suggested to present the findings in a more concise and focused manner, avoiding extraneous details unrelated to the core conclusions.

Response:

We agree that the Conclusions section could be more concise. The text was modified to focus on the major findings of the study and potential future perspectives of the research.